# Trp Fluorescence Redshift during HDL Apolipoprotein Denaturation Is Increased in Patients with Coronary Syndrome in Acute Phase: A New Assay to Evaluate HDL Stability

**DOI:** 10.3390/ijms22157819

**Published:** 2021-07-22

**Authors:** Victoria López-Olmos, María Luna-Luna, Elizabeth Carreón-Torres, Héctor González-Pacheco, Rocío Bautista-Pérez, Rosalinda Posadas-Sánchez, José Manuel Fragoso, Gilberto Vargas-Alarcón, Óscar Pérez-Méndez

**Affiliations:** 1Department of Molecular Biology, Instituto Nacional de Cardiología Ignacio Chávez, Mexico City 14080, Mexico; vickyolmosunam@hotmail.com (V.L.-O.); mjluna.qfb@gmail.com (M.L.-L.); elizact73@gmail.com (E.C.-T.); rociobtst@yahoo.com (R.B.-P.); mfragoso1275@yahoo.com.mx (J.M.F.); gvargas63@yahoo.com (G.V.-A.); 2Emergency Department, Instituto Nacional de Cardiología Ignacio Chávez, Mexico City 14080, Mexico; hectorglezp@hotmail.com; 3Department of Endocrinology, Instituto Nacional de Cardiología Ignacio Chávez, Mexico City 14080, Mexico; rossy_posadas_s@yahoo.it; 4School of Engineering and Sciences Campus CDMX, Instututo Tecnólogico y de Estudios Superiores de Monterrey, Mexico City 14380, Mexico

**Keywords:** Trp fluorescence, HDL stability, atherosclerosis, protein structure, coronary heart disease

## Abstract

High-density lipoproteins’ (HDL) stability is a determinant of their residence times in plasma and consequently an important parameter that influences the beneficial properties of these lipoproteins. Since there are no accessible procedures for this purpose, here, we describe the methodological conditions to assess the stability of the HDL based on the redshift of the fluorescence spectrum of tryptophans contained in the structure of HDL-apolipoproteins during incubation with urea 8M. Along the HDL denaturation kinetics, the main variations of fluorescence were observed at the wavelengths of 330, 344, and 365 nm at room temperature. Therefore, HDL denaturation was estimated using the tryptophan (Trp)-ratio of fluorescence intensity ($r_{fi}$) at such wavelengths. By setting 100% of the measurable denaturation at 26 h, HDL reached 50% after 8 h of incubation with urea. Then, for further analyses we determined the percentage of HDL denaturation at 8 h as an estimation of the stability of these lipoproteins. To explore the potential usefulness of this test, we analyzed the stability of HDL isolated from the plasma of 24 patients diagnosed with acute coronary syndrome (ACS). These HDL presented significantly higher percentages of denaturation (64.9% (58.7–78.4)) than HDLs of healthy individuals (23.3% (20.3-27.0)). These results indicate that HDL in ACS are less stable than in control subjects. Moreover, the percentage of denaturation of HDL correlated with body mass index and aspartate transaminase plasma activity. Furthermore, apo-I, HDL-cholesterol, HDL-triglycerides, and apo A-I-to-triglycerides ratio correlated with the percentage of HDL denaturation, suggesting that the lipoprotein composition is a main determinant of HDL stability. Finally, the percentage of HDL denaturation is the parameter that predicted the presence of ACS as determined by a machine learning procedure and logistic regression analysis. In conclusion, we established the methodological conditions to assess the stability of HDL by a fluorescence-based method that merits exploration in prospective studies for evaluating the coronary artery disease risk.

## 1. Introduction

High-density lipoproteins (HDL) prevent the atheroma formation in the arteries mainly by promoting cholesterol transport from the tissues to the liver [1]. In addition, HDL have vasodilator effects since they induce nitric oxide production and proliferation of endothelial cells, decreasing vascular inflammation and oxidative damage [2]. The protective functions exerted by HDL can be impaired in the setting of certain diseases (diabetes, metabolic syndrome, chronic kidney disease, chronic inflammation, and coronary heart disease); such HDL impairment has been associated with protein alterations (post- translational modifications), and HDL proteomics modifications [3,4]. High apolipoprotein (apo) A-I catabolic rates are a common feature in such physiopathological situations [5,6,7], suggesting instability of the HDL particles that facilitate the dissociation of the apo A-I and further clearance by the kidney [5,8,9]. As a consequence, the dysfunctional HDL may be related to the instability of these lipoproteins. Therefore, a measure of the HDL stability is a potential tool to better understand the role of these lipoproteins in coronary artery disease (CAD) and its clinical manifestation, the acute coronary syndrome (ACS).

In terms of structure, the stability of HDL depends upon apolipoprotein A-I (apo A-I), which represents 70% of the total protein content within these lipoproteins, and to a lesser extent, apo A-II, apo A-IV, and apo-E [1,9,10]. Mature apo A-I is a 243-amino-acid protein, including four residues of tryptophan (Trp); the protein is assembled through interactions of hydrophobic amino acids with the acyl residues of phospholipids, while its charged amino acids interact with the aqueous environment and polar residues of phospholipids [11]. Apo A-II represents between 15% to 20% of the apolipoproteins and lacks tryptophan residues [12]; the remaining apolipoproteins are presented in low proportion. When HDL are treated with a chaotropic agent such as urea or guanidinium hydrochloride, the apolipoproteins become unfolded and expose their hydrophobic residues to the aqueous environment. Then, in the presence of chaotropic agents, the intrinsic fluorescence of Trp that is typically embedded in the hydrophobic phase of intact HDL is redshifted because of the dipole moment increase in the surrounding environment of the amino acid [13,14]; under these conditions, how fast the Trp fluorescence of HDL proteins is redshifted depends upon the strength of interaction between apolipoproteins and lipids. In other words, the kinetics of redshift of HDL proteins may provide information about the stability of the HDL particles. Therefore, the objectives of the present study were to establish the experimental conditions to determine the stability of the HDL based on the kinetics of Trp fluorescence redshift induced by urea and to explore whether this method is potentially useful to discriminate patients with ACS from individuals without coronary artery disease.

## 2. Results

### 2.1. Definition of The Meaningful Parameters to Estimate HDL Stability

The initial assays to establish the conditions for the new test using the redshift of Trp fluorescence emission of HDL proteins; we used HDL isolated from plasma of six apparently healthy volunteers. Trp spectra had a satisfactory definition with low noise from 20 to 50 μg/mL of HDL protein (data not shown). Therefore, we selected a concentration of 30 μg/mL which is above the minimal protein concentration to obtain an acceptable fluorescence spectrum, and low enough to use a small amount of the plasma sample.

Under native conditions, the Trp fluorescence emission spectra of HDL proteins were characterized for two peaks, one at λ = 330 nm (Emλ330) and one at λ = 344 nm (Emλ344) (Figure 1a). Under denaturing conditions with urea 8 M (Figure 1b), the fluorescence intensity at λ = 330 nm gradually decreased over time; in parallel, Emλ344 and the fluorescence at λ = 365 nm (Emλ365) drastically increased (Figure 1b).

Based on these observations, we assumed that the Emλ330 corresponds to the Trp more deeply embedded in the hydrophobic phase of the lipoprotein, whereas Emλ344 and Emλ365 are the result of the exposure of Trp residues to the aqueous phase due to the denaturing process. Therefore, we defined the ratio of fluorescence intensities ($r_{fi}$) as: $r_{fi}=\frac{\left(\mathrm{Em}\lambda 344+\;\mathrm{Em}\lambda 365\right)}{\mathrm{Em}\lambda 330}$. This parameter estimates the degree of denaturation of HDL particles; the higher the $r_{fi}$, the greater the degree of denaturation. A representative time-course of $r_{fi}$ in the presence of urea 8M is presented in Figure 2a. After 26 h in 8 M of urea, the fluorescence spectra maintained the same shape, but the fluorescence intensity gradually decreased (data not shown). Therefore, we defined that HDL protein at 26 h reached 100% of the measurable denaturation. From this assumption, we established that the value of the $r_{fi}$ at basal conditions (t = 0 h) was equivalent to 0% of denaturation ($r_{fi}{}_{0}=0\%$) and the $r_{fi}$ after 26 h of incubation with urea 8 M represented 100% of denaturation ($r_{fi}{}_{26}=100\%$). The intermediate data were then converted to percentage of denaturation and plotted vs. time (Figure 2b); each data set (*n* = 6) were adjusted to a sigmoid curve $\left(y=\frac{1}{1+e^{-x}}\right)$, where $\;y=r_{fi}$ is expressed as percentage of denaturation and x=time (h). By this, we defined the mean time at which the $r_{fi}$ reached 50% of the maximal value ($r_{fi}{}_{26}$), and therefore considered it as 50% of the denaturation time (DT50). We found that the DT50 was 8.0 ± 0.5 h for this initial group of healthy individuals. Therefore, at 8 h, variations of the percentage of HDL protein denaturation above and below 50% could be detected with important sensibility. Consequently, for further analyses to estimate the stability of HDL we fixed the 8 h of incubation to the time to determine the percentage of denaturation using the $r_{fi}$. Under these conditions, the intra and inter-assay variation coefficients of the test were 8.3% and 8.9%, respectively.

### 2.2. Analysis of HDL Integrity during Denaturalization

The protein and cholesterol associated with the HDL separated by PAGE in native conditions are shown in Figure 3. Surprisingly, most of the HDL proteins remained associated to cholesterol in the presence of urea 8 M (Figure 3a,b) and just a fraction of the proteins detached from the lipoprotein; free HDL proteins were not detected in electrophoresis at 0 h, whereas these free proteins represented 11.0%, 12.9%, 16.0% and 19.6%, and 21.0% and 32.4% of the total HDL proteins, at 2 h, 4 h, 8 h, 12 h, and 26 h, respectively, as determined by the area under the curve of the densitogram (Figure 3c). Concerning the non-denaturized HDL, the size distribution of these particles was not notably modified along the incubation time as observed in the densitograms (Figure 3c,d).

### 2.3. Determination of the Percentage of HDL Proteins in Patients with ACS and Controls

After establishing the conditions for this new test to estimate the percentage of denaturation as a marker of HDL stability, we investigated the potential usefulness of the assay to discriminate patients with ACS—a clinical manifestation of coronary artery disease. Plasma samples from patients were obtained within 24 h after the beginning of the ACS symptoms.

Anthropometric and biochemical parameters of the included subjects are shown in Table 1. The body mass index (BMI) in ACS patients was significantly higher than that of control individuals (*p* < 0.05). The age, sex, systolic and diastolic blood pressure did not show significant differences between groups (Table 1). Moreover, there was a significant 50% higher alcohol consumption in clinically healthy individuals than ACS patients (*p* < 0.05). Patients were receiving pharmacological treatments, particularly statins. Consequently, the cholesterol and LDL-cholesterol plasma levels were significantly lower in ACS patients than in the control subjects (Table 1). Values of fasting plasma levels of glucose, uric acid, alanine aminotransferase (ALT), and aspartate aminotransferase (AST) plasma activities were significantly higher in patients than in controls (*p* < 0.05). Apo A-I, HDL-cholesterol, and HDL-phospholipids were 31%, 34%, and 27% lower in patients than in controls, whereas HDL-triglycerides were about 40% higher in patients than in controls (Table 1).

Concerning HDL stability, the results are shown in Figure 4. The percentage of HDL protein denaturation in ACS patients was about three times higher that of the healthy controls (64.9% (58.7–78.4%) vs. 23.3% (20.3–27.0%), respectively, *p* < 0.05). These results indicated that HDL were less stable in ACS patients than in healthy individuals.

To determine whether the percentage of denaturation of HDL could be proposed as an additional tool to evaluate the cardiovascular risk, we first performed a Mann–Whitney U test to compare the percentage of denaturation of HDL between ACS and control groups (Figure 4). Since the median of patients and controls were drastically different, we further analyzed data by a prediction model in machine learning as described in the Materials and Methods Section. The effectivity of the model in the training database (75% of the subjects) was 87.5% for the prediction model in the machine learning, when BMI, alcohol intake, smoking habit, SBP, glucose, total cholesterol, triglycerides, creatinine, AST, and the percentage of HDL denaturation were included. In contrast, the effectivity increased up to 97% in the training database, and it was 100% in the verification database when only the percentage of HDL denaturalization was considered. We further performed the logistic regression analysis including the subjects considered in the training database (*n* = 33); under these conditions, the calculated OR was 4.7965, with a confidence interval (1.226–8.367), and *p* = 0.008.

We further explored which biochemical parameters may determine the stability of HDL as estimated by this method. For this, we initially performed a Spearman analysis, and the significant correlations are presented in Table 2. Glucose, total cholesterol, LDL-cholesterol and AST significantly correlated with the percentage of denaturation of HDL at 8 h.

Concerning the potential contribution of the lipid content to the stability of HDL, we observed a strong correlation of all the components of these lipoproteins with the percentage of denaturation (Table 2). Apo A-I-to-HDL-cholesterol ratio positively correlated with the percentage of denaturation. In contrast, the correlation of apo A-I-to-HDL-triglycerides ratio with HDL denaturation was negative (Table 2). It is important to emphasize that correlations were enhanced by the strong segregation of control and patient groups as observed in the dispersion plots for regression analysis (Appendix A).

To statistically consider the effect of statins on the denaturation of HDL, we performed the partial correlation introducing statins intake as covariate; in this analysis, only BMI (*r* = 0.323, *p* = 0.037) and HDL-cholesterol (*r* = −0.311, *p* = 0.045) remained significantly correlated with the percentage of denaturation of HDL at 8 h. Apo A-I and other HDL-lipids did not further correlate after the adjustment by the intake of antidyslipidemic drugs (data not shown).

As an approach to the HDL lipid composition, we calculated the apo A-I/HDL-lipids ratio; remarkably, the apo A-I-to-HDL-triglycerides ratio in control subjects was about twice (8.1 ± 1.5 mg of apo A-I/mg of triglycerides) the value determined in patients (4.4 ± 1.9 mg of apo A-I/mg of triglycerides, *p* < 0.001). The proportion of apo A-I/HDL-cholesterol and apo A-I/HDL-phospholipids were similar between patients and controls (3.3 ± 0.6 vs. 3.0 ± 0.3 mg of apo A-I/mg of cholesterol, and 1.4 ± 0.2 vs. 1.4±0.1 mg of apo A-I/mg of phospholipids, respectively, *p* > 0.05 for both). Therefore, HDL particles were only enriched with triglycerides in patients with ACS.

Considering that cholesterol is one of the components of HDL that was importantly associated with the percentage of denaturation, we explored whether at a same level of plasma HDL-cholesterol denaturation was similar. For this, we were able to match nine patients with nine control subjects at ± 1mg/dL of HDL-cholesterol; Figure 5 depicts these subgroups. HDL-cholesterol for controls and patients were 42.9 (37.5–48.1) vs. 41.8 (36.6–49.8) mg/dL, respectively, *p* > 0.05. Interestingly, the percentage of HDL denaturation in the subgroups (21.8% (20.6–25.6%) vs. 64.2% (52.6–71.8%) for controls and patients respectively, *p* < 0.001) was comparable with the initial groups described above.

## 3. Discussion

The present study describes a method to estimate the stability of HDL based on the denaturation of their apolipoproteins at a fixed time; the results are expressed as a percentage of denaturation considering a maximal value of $r_{fi}{}_{26}$. This new parameter was about three times higher in patients with ACS than in control subjects, suggesting that the percentage of denaturation may be a good marker of cardiovascular disease.

The apolipoproteins that became detached from lipids after 26 h of incubation with urea only reached 32.4% of the total HDL proteins as estimated by electrophoresis in non-denaturing conditions. Surprisingly, most of the HDL protein remained associated with the lipids after this period. In accordance with this observation, the fluorescence at λ = 330 nm did not completely disappear, it only decreased in the presence of urea during the time of follow-up. Moreover, the detached apolipoprotein fraction is the most susceptible to expose its Trp residues to the aqueous environment, thus redshifting their λ_max_ of fluorescence emission. The significant increases in the peaks at λ = 344 and λ = 365 nm were mainly due to the apolipoprotein fraction that became separated from the HDL particle. This fact explains why we observed three different peaks of fluorescence emission [15]. Therefore, the fluorescence emission spectra of HDL along the incubation with urea were a composite of distinct molecules of protein exposing their Trp residues to the aqueous media in different degree. Therefore, we included the relative fluoresce at three wavelengths to calculate the $r_{fi}$ of those that systematically changed along the incubation time with urea in order to obtain a more sensitive method.

Despite the partial denaturation of HDL, we fixed the maximal time of incubation at 26 h because the intrinsic fluorescence of Trp decreased during this time; therefore, longer incubations would implicate a loss of sensitivity and accuracy of the test. In addition, longer incubation times were not associated with significant increases in the $r_{fi}$ (data not shown). Based on such observations, we assigned the value of $r_{fi}{}_{26}$. as the maximal detectable denaturation (100%).

The observed partial dissociation of apolipoproteins from HDL is also of interest in the context of early reports concerning apo A-I catabolism [16]; in those studies, it was demonstrated that there was an exchangeable fraction of apo A-I that was not tightly associated to HDL. The amount of apo A-I fraction that became detached from the lipoprotein was higher and also faster catabolized in hypoalphalipoproteinemic subjects than in normolipidemic subjects. Likely, the integrity of HDL is inversely correlated with the catabolism of apo A-I [16]. Accordingly, HDL-cholesterol was the only one component of HDL that correlated with the percentage of denaturation in the analysis after adjustment by statins intake. Nevertheless, when we compared patients and controls paired by HDL-cholesterol, the percentages of HDL denaturation were very important for the entire group. Therefore, HDL-cholesterol plasma level may be a surrogate of HDL stability, but it is not a determinant of their percentage of denaturation.

Once the experimental conditions were established, we analyzed the stability of HDL obtained from patients with diagnosis of ACS to explore the potential usefulness of the test. Our results demonstrated a huge difference concerning the percentage of denaturation between patients and controls. To gain more insight into the association of the new parameter as an independent predictor of ACS, and considering the limited number of data, we perform an initial analysis using a machine learning methodology. The algorithm initially includes a subgroup of individuals named training database. Then, the predictor parameters derived from this subset of individuals are considered in the remaining group of subjects, named the verification database, to test whether these parameters can predict the ACS incidence. The method considered the entire database to look for the potential factors that would predict the presence of ACS. The effectivity of HDL denaturation plus the traditional risk factors of coronary artery disease predicted only partially the presence of ACS. However, the model reached an effectivity of 100% when the model included only the percentage of HDL denaturation. With this antecedent, we performed the logistic regression analysis in the training database, including only the percentage of HDL denaturation as an independent variable. The results showed a strong inverse association of the percentage of HDL denaturation with the presence of ACS.

The mechanisms that lead to HDL stability should be linked to the intrinsic characteristics of these lipoproteins. In this context, AST plasma activity and BMI independently explained the percentage of HDL denaturation in the studied population. The former may be related to hepatic function; considering that HDL are secreted by the liver, a dysfunction of this organ, i.e., fatty liver, may affect the HDL assembly and lipid composition, generating less stable particles [17]. Furthermore, the BMI is a known CAD risk factor, generally associated with low HDL levels that become enriched with triglycerides in the onset of metabolic syndrome [18]; this enrichment with triglycerides may destabilize the structure of HDL. Accordingly, HDL-triglycerides’ plasma concentration positively correlated with the percentage of HDL denaturation. Nevertheless, the plasma concentration of HDL-triglycerides does not clearly reflect the content of this lipid within HDL particles. A better approximation to HDL composition is the ratio of apo A-I-to-HDL-lipids. Consequently, apo A-I-to-HDL-triglycerides ratio showed a strong negative correlation with the percentage of denaturation; since the higher the ratio the lower the content of triglycerides, then these results support the idea that the triglycerides content of HDL favors the denaturation of the lipoprotein.

Furthermore, apo A-I, HDL-cholesterol, and HDL-phospholipids plasma concentrations were lower in patients than in controls, and negatively correlated with the percentage of denaturation. These data suggest that patients had a lower number of HDL particles than controls, and that such a low number of particles is likely a determinant for the percentage of denaturation. It should be considered that correlations mentioned above could be also influenced by the data segregation in clusters, a consequence of the very important difference of the percentage of HDL denaturation between controls and ACS patients. Importantly, most of the correlations did not remain significant after adjustment of the analysis by statins intake; this statistical effect was mainly derived from the fact that only the patients were taking antidyslipidemic drugs.

Finally, we recognize that the number of studied subjects is small to reach definite conclusions. In this regard, we intended to set a precedent for the potential usefulness of our method for future prospective studies that will assist in evaluating the risk of coronary artery disease. Moreover, patients were in the acute phase of the coronary syndrome; during this period, there are important alterations of HDL structure [3,4,19,20,21] that may have altered the stability of HDL. Additionally, patients were receiving different drugs at the time of hospitalization, including statins that may alter the metabolism of HDL and increase the plasma activity of transaminases. All these issues may have influenced the results and merit investigation in further studies.

## 4. Materials and Methods

### 4.1. Study Population

To define the experimental conditions that determine the stability of HDL via the redshift of Trp fluorescence emission, we recruited, from the blood bank of the National Institute of Cardiology, 6 clinically heathy volunteers (2 men and 4 women), without personal and familiar history of coronary artery disease (CAD).

To explore the potential usefulness of the percentage of denaturation of HDL, we further enrolled 24 patients (19 men and 5 women) diagnosed with acute coronary syndrome (ACS) according to standard definitions by the American College of Cardiology [22]. The medical history was reviewed and anthropometric parameters, blood pressure, pharmacological treatment, alcohol consumption, active tobacco habit, and biochemical parameters were registered. This part of the study was nested in the cohort study from the Coronary Care Unit of the National Institute of Cardiology Ignacio Chávez in México City [21]. As the control group, we recruited 20 clinically healthy individuals, 12 men and 8 women from the GEA cohort [23], matched by age with the ACS patients. The inclusion criteria for control individuals were a BMI < 27 kg/m^2^, coronary artery calcium (CAC) score = 0, and with a normal lipid profile (total cholesterol ≤ 200 mg/dL, triglycerides ≤ 150 mg/dL, HDL-cholesterol ≥ 40 mg/dL). The study was in agreement with the guidelines of the Helsinki declaration and approved by the Ethics Committee of the National Institute of Cardiology Ignacio Chávez. Informed consent was obtained from all subjects involved in the study.

### 4.2. Biochemical Analyses

Venous blood samples were collected after a 12 h fast in tubes with EDTA and centrifuged for 15 min at 1300× *g*. Plasma levels of total cholesterol, HDL-cholesterol, triglycerides, AST, ALT, and glucose were measured in fresh samples through standardized enzymatic/colorimetric methods as previously reported [24]. The control plasmas were stored at −70 °C in a similar period as patient samples. The blood samples from ACS patients were drawn within the first 24 h of admission to the coronary care unit and the plasma did not exceed more than one month of freezing at −70 °C.

### 4.3. Isolation of HDL

HDL were isolated from 2 mL of frozen plasma by differential ultracentrifugation (table centrifuge, Beckman optima TLX, Indianapolis, IN, USA) as previously described [8] and dialyzed against NaCl 150mM, Na_2_HPO_4_/NaH_2_PO_4_ 10mM buffer, (PBS) pH 7.4. Protein concentration was determined by Lowry’s method [25]. The albumin content in the isolated HDL was determined by a 3–30% polyacrylamide gradient gel electrophoresis in native conditions as previously described [26,27,28]. The albumin content in the HDL for denaturation studies was <3% of the total protein.

### 4.4. Redshift of Trp Fluorescence during HDL Denaturation

The denaturation of HDL was determined by the Trp fluorescence spectra along the time. The denaturation experiments were carried out with isolated HDL at a final concentration of 30 μg/mL, in PBS-8 M urea, pH 8.1 [29]. Control samples were incubated in the absence of urea. The Trp intrinsic fluorescence spectra were obtained with a LS55 PerkinElmer fluorescence spectrophotometer (BUCKS, UK) at room temperature, with an excitation wavelength (λ_ex_) of 280 nm and emission wavelength (λ_em_) interval from 300 to 400 nm, and with a bandwidth of 5.5 nm for both. For every sample, spectra were recorded at initial time (t = 0 h), at 8 h, or every hour during 12 h for some experiments. A final fluorescence spectrum was recorded at 26 h, which was set as the time at which 100% of the measurable denaturation of HDL was reached.

The ratio of fluorescence peaks at different λ_em_ was assayed to define the most reproducible and accurate value for assessing the percentage of denaturation. For some experiments, the curves of denaturation along the time were adjusted to a sigmoid function using Sigma Plot software, where the inflection point was defined as the 50% of HDL protein denaturation. The parameters obtained from spectra were analyzed to define the mean time at which HDL proteins were denatured.

The intra- and inter-assay coefficients of variability (CV) of the test were assessed with a plasma pool of four volunteers from the first stage of the study; the intra-assay CV was obtained from ten replications of the same sample, and repeated in 3 different days, whereas the inter-assay CV was assessed by 10 independent repetitions along 6 weeks.

### 4.5. Analysis of HDL Integrity during Denaturalization

To assess the integrity of the HDL particle along the incubation time with urea 8 M, we first exchanged the buffer containing urea by 0.09 M Tris/0.08 M boric acid/3 mM EDTA (TBE) buffer, pH 8.4 using centrifugal filters with a nominal molecular weight limit of 10 kD (Amicon Ultra 0.5 mL Centrifugal Filters, MERCK KGAA, Darmstad, Germany). Buffer exchange was completed within the next 45 min after the end of incubation periods. Then, we determined the HDL size distribution in polyacrylamide gradient gels under native conditions as previously described [26,27,28]. For this, electrophoresis gels were first enzymatically stained for cholesterol, scanned, and further stained with Coomassie blue R-250 for proteins as we have previously described [26]. Densitograms of the scanned gels were obtained with the Quantity One, version 4.5, software (Bio-Rad, Hercules, CA, USA). The addition of all the areas under the curves in the densitograms, including the peaks not associated with cholesterol, stained with Coomassie blue R-250, were considered as 100% of HDL protein. With this reference, the percentage of the detached protein from HDL was calculated using the area under the corresponding peak.

### 4.6. Statistical Analysis

The distribution of data was screened by the Shapiro–Wilk test. Student’s paired *t*-test (normally distributed data) and Mann–Whitney U test (non-normally distributed data) were used to compare groups, whereas the categorical variables were compared with a *χ*^2^ test. Values with normal distribution were expressed as mean ± standard deviation (SD)and non-normal distributed data were expressed as median and interquartile range. These analyses were performed with the Statistical Package for the Social Sciences (SPSS), version 22 software (International Business Machines Corp, Armonk, NY, USA).

We further explored the possibility that the percentage of denaturation of HDL predicts the presence of ACS. For this, the proposed model initially considered the complete data from all the individuals in the study with ACS as the dependent variable. Then, data were standardized and further requested in 75% for the training database and 35% for the verification database. In order to remove characteristics with low weight, we used the Recursive Feature Elimination algorithm with the scikit-learn module, machine learning in Phyton. To test the model’s effectiveness, we performed the test of successes or effectiveness in the training base. Finally, a logistic regression model was constructed with Statsmodels y Sklearn linear modules of Phyton. Statistical significance was considered with a *p* value < 0.05.

## 5. Conclusions

We described a method to estimate the stability of HDL based on the redshift of the Trp fluorescence emission spectrum during the denaturation of their apolipoproteins in urea 8 M. Under the conditions that we described, this percentage of denaturation of HDL was about three times higher in patients with ACS than in control subjects, suggesting that it may be a good marker of cardiovascular disease.

## Figures and Tables

**Figure 1 ijms-22-07819-f001:**
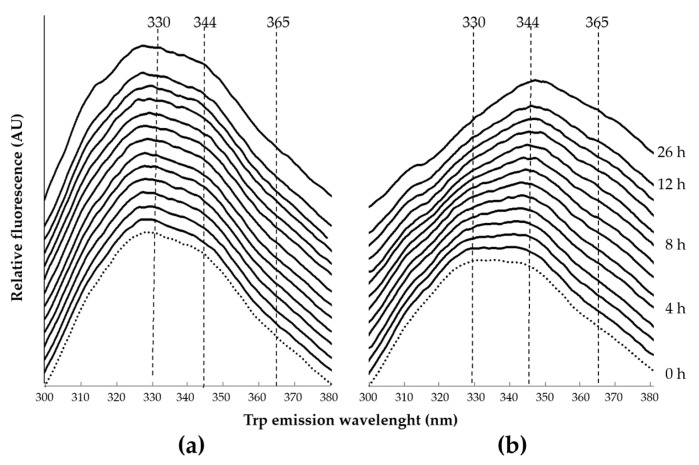
Trp fluorescence emission spectra of HDL of a representative healthy subject. The fluorescence intensity of HDL without urea (**a**) and with 8M urea (**b**) are shown. Spectra were recorded every hour using 30 µg/mL of HDL protein concentration, normalized, and depicted chronologically from the bottom to the top. The wavelengths considered to estimate the denaturation of HDL-apolipoproteins are indicated by the vertical dotted lines. The excitation wavelength was set at 280 nm. AU, arbitrary units.

**Figure 2 ijms-22-07819-f002:**
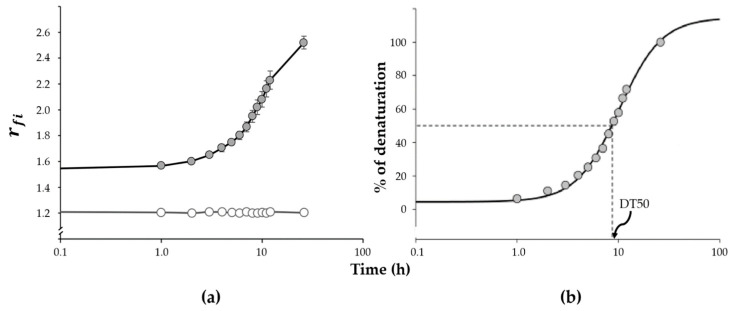
Effect of urea on the denaturation of HDL-apolipoproteins. (**a**) The fluorescence in healthy subjects (*n* = 6) was monitored as described in Figure 1: without urea (open circles) or 8 M urea activity (gray circles) and represented as $r_{fi}\;$(see the text for details). (**b**) Data set were adjusted to a sigmoid curve $\left(y=\frac{1}{1+e^{-x}}\right)\;$ using the SigmaPlot software in order to define the mean time at which 50% of the HDL-proteins were denatured (DT50), establishing that the value of the $r_{fi}\;$ at basal conditions (t = 0 h) was equivalent to 0% of denaturation and the ratio after 26 h of incubation with urea 8 M represented 100% of denaturation.

**Figure 3 ijms-22-07819-f003:**
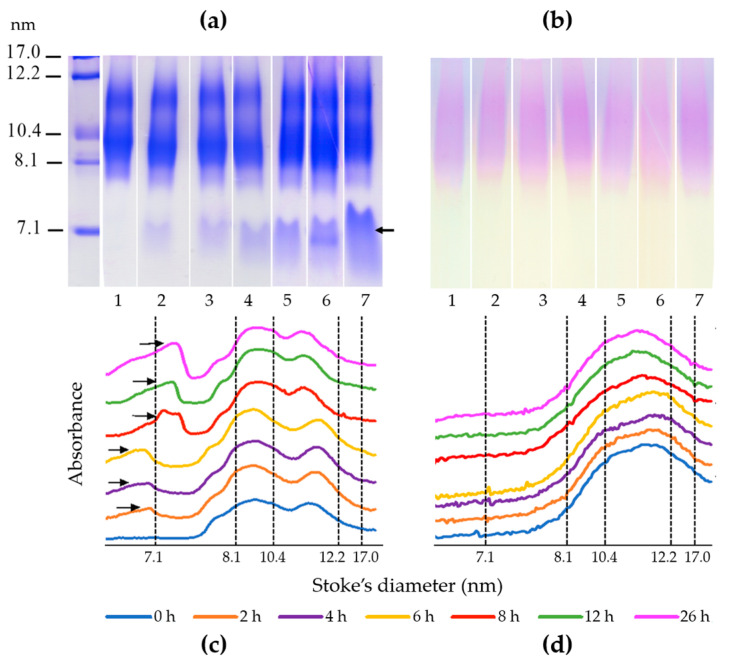
Electrophoretic analysis of HDL denaturation in healthy subjects. HDL (pooled plasma) were isolated by ultracentrifugation and incubated with urea 8 M at different times. The urea was removed and denatured HDL were separated by their size in a native 3–30% polyacrylamide gel electrophoresis; (**a**) stained with Coomassie blue for protein and (**b**) enzymatically for cholesterol. Gels were first stained for cholesterol, destained, and stained further for proteins. Denaturation of HDL was evaluated at 0 h (lanes 1), 2 h (lanes 2), 4 h (lanes 3), 6 h (lanes 4), 8 h (lines 5), 12 h (lanes 6), and 26 h (lanes 7). (**c**) Densitograms of the gels stained for proteins and (**d**) stained for cholesterol at different times were normalized and depicted using arbitrary absorbance units on the vertical axis. Arrows indicate the fraction of proteins not associated to HDL, which gradually increased along the time of incubation with urea, whereas the profile of HDL-cholesterol remained essentially unaltered.

**Figure 4 ijms-22-07819-f004:**
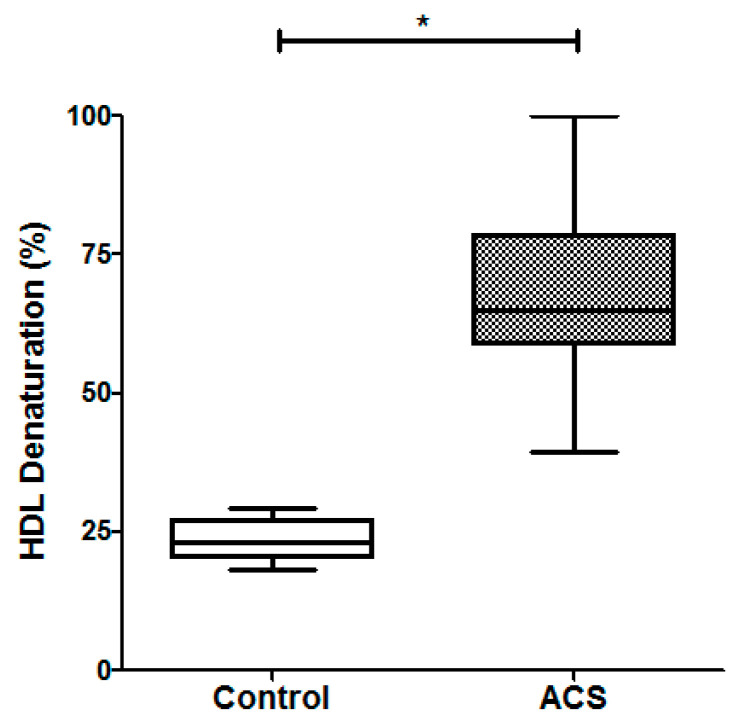
Percentage of HDL denaturation after 8 h of incubation with urea 8 M, in ACS patients (*n* = 24) and controls (*n* = 20). Data are represented as the median (line horizontal) and interquartile range (boxes). Mann–Whitney U test. * *p* < 0.05.

**Figure 5 ijms-22-07819-f005:**
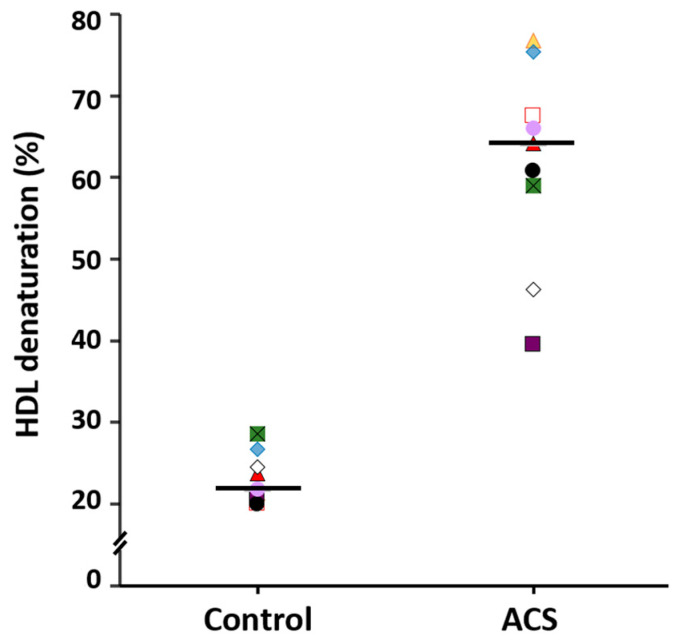
Percentage of HDL denaturation after 8 h of incubation with urea 8 M, in ACS patients (*n* = 9) and controls (*n* = 9) matched by HDL-cholesterol plasma levels. Same symbols represent the paired controls and ACS patients. Horizontal bar represents the median of the subgroup.

**Table 1 ijms-22-07819-t001:** Anthropometric characteristics, biochemical data, and pharmacological treatment of the included subjects.

Parameter	Control Subjects (*n* = 20)	ACS Patients (*n* = 24)	* *p* Value
Age (years)	57 ± 1.87	55 ± 2.57	NS ^a^
Sex, *n* [M (%)/F (%)]	12 (60)/8 (40)	19(79.16)/5(20.83)	NS ^c^
Body mass index (kg/m^2^)	25.28 ± 2.13	27.81 ± 5.94	0.034 ^b^
Systolic blood pressure (mmHg)	112.6 ± 2.18	121.5 ± 3.64	NS ^a^
Diastolic blood pressure (mmHg)	71 [68.12–75.37]	70 [67.75–74.00]	NS ^b^
Alcohol drink, *n* (%)	16 (80)	7 (29)	0.001 ^c^
Active smokers, *n* (%)	4 (20)	10 (41.66)	NS ^c^
Statin use, *n* (%)	0 (0)	21 (88)	0.000 ^c^
β-blockers use, *n* (%)	0 (0)	13 (54)	0.000 ^c^
Calcium inhibitor antagonists use, *n* (%)	0 (0)	5 (20.83)	0.000 ^c^
Glucose (mg/dL)	92.95 [86.2–95.17]	100.80 [94.07–138.98]	0.000 ^b^
Total cholesterol (mg/dL)	179.78 ± 5.20	135.94 ± 7.08	0.000 ^a^
LDL-cholesterol (mg/dL)	106.11 ± 4.96	81.11 ± 5.56	0.002 ^a^
Triglycerides (mg/dL)	128.09 ± 9.32	142.35 ± 11.32	NS ^a^
HDL-cholesterol (mg/dL)	53.17 ± 2.49	34.60 ± 2.06	0.000 ^a^
HDL-triglycerides	19.7 ± 3.9	27.5 ± 10.5	0.003 ^a^
HDL-phospholipids	109.4 ± 20.3	78.8 ± 20.4	0.000 ^a^
Apo A-I	154.8 ± 19.9	106.4 ± 27.6	0.000 ^a^
ALT (IU/L)	18.65 [12.93–24.17]	30.10 [16.05–67.65]	0.022 ^b^
AST (IU/L)	20.90 [17.83–23.70]	32.90 [22.73–112.17]	0.000 ^b^
Uric acid (mg/dL)	5.48 ± 0.28	6.90 ± 0.44	0.013 ^a^
Creatinine (mg/dL)	0.921 ± 0.044	1.16 ± 0.18	NS ^a^

Data are expressed as mean ± SD, median (interquartile range) or percentage. ^a^ Student’s t-test, ^b^ Mann–Whitney U, and ^c^ χ^2^ test analysis. * *p* value for the comparison between patients and control subjects. ACS: acute coronary syndrome. ALT: alanine aminotransferase. AST: aspartate aminotransferase. NS: not statistically significant.

**Table 2 ijms-22-07819-t002:** Correlation analysis between the percentage of HDL denaturation at 8 h of incubation with urea 8 M and biochemical parameters.

Parameter	*r*	*p*
BMI	0.312	0.044
Glucose	0.501	0.001
Total cholesterol	−0.559	0.000
LDL-cholesterol	−0.382	0.013
AST	0.406	0.008
Uric acid	0.316	0.044
HDL-cholesterol	−0.684	0.000
HDL-triglycerides	0.450	0.002
HDL-phospholipids	−0.673	0.000
Apo A-I	−0.706	0.000
Apo A-I-to-HDL-cholesterol ratio	0.311	0.045
Apo A-I-to-HDL-triglycerides ratio	−0.666	0.000
Apo A-I-to-HDL-phospholipids ratio	0.040	0.802

Spearman analysis, *n* = 44. To the exception of apo A-I-to-HDL-phospholipids ratio, only the significant correlations are shown. BMI: body mass index. AST: aspartate transaminase.

## Data Availability

The data presented in this study are available on request from the corresponding author. The data are not publicly available for protection of personal information.

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
