# Peer review of "Trp Fluorescence Redshift during HDL Apolipoprotein Denaturation Is Increased in Patients with Coronary Syndrome in Acute Phase: A New Assay to Evaluate HDL Stability"

_ijms, 2021, doi:10.3390/ijms22157819_

Round 1

Reviewer 1 Report

Title: Trp-fluorescence redshift during HDL apolipoprotein denaturation is increased in patients with acute coronary syndrome: a new assay to evaluate HDL stability

Authors: Victoria Lopez-Olmos et al.

General Comment:  This is a well written manuscript.  The experiments were well executed.  The statistical analyses are well performed.  However, the usefulness of this assay to predict ACS is debatable.  It is a new assay to evaluate the HDL stability.

Specific comment:

The statement in Introduction on the rational of why the “the kinetics of redshift of HDL Proteins may provide information about the stability of HDL” needs to be made clearer and should provide some references to support the rational.  Please rewrite this part.

Author Response

Response to Reviewer 1 Comments

General Comment: This is a well written manuscript. The experiments were well executed. The statistical analyses are well performed. However, the usefulness of this assay to predict ACS is debatable. It is a new assay to evaluate the HDL stability.

Response: We would like to thank the Reviewer for the time spent for this evaluation and for the positive appreciation of our study. The modifications made in the manuscript were highlighted in the corrected version of the manuscript.

Specific comment: The statement in Introduction on the rational of why the “the kinetics of redshift of HDL Proteins may provide information about the stability of HDL” needs to be made clearer and should provide some references to support the rational. Please rewrite this part.

Response: Following the Reviewer’s recommendation, we have included in the introduction section, page 2, second paragraph,the following modified sentences:

Then, in the presence of chaotropic agents,the intrinsic fluorescence of Trp that is typically embedded in the hydrophobic phase of intact HDL is redshifted because of the dipole moment increase in the surrounding environment of the amino acid [13,14]; under these conditions, how fast the Trpfluorescence of HDL-proteins is redshiftedd epends upon the strength of interaction between apolipoproteins and lipids. In other words, the kinetics of redshiftof HDL-proteins may provide information about the stability of the HDL particles.

We hope that this modification makes clearer the concept.

Reviewer 2 Report

The authors report a method to measure the propensity of HDL to denaturation and evaluate HDL stability. The authors propose that this method can be used to predict the presence of ACS. However, this cannot be proved by the provided analyses.
The method should be performed in HDL containing different concentrations of cholesterol and apoA-I both from healthy controls and ACS subjects. 
1) What is the effect of cholesterol (or phospholipds and other lipids) levels and apoA-I (or apoA-II and other protein) levels in HDL denaturation and stability?
2) Do HDL from ACS subjects with similar cholesterol levels as controls display difference in denaturation?
3) Was the plasma of ACS subjects collected during acute or stable phase?

Author Response

Response to Reviewer 2 Comments

General Comment: The authors report a method to measure the propensity of HDL to denaturation and evaluate HDL stability. The authors propose that this method can be used to predict the presence of ACS. However, this cannot be proved by the provided analyses.The method should be performed in HDL containing different concentrations of cholesterol and apoA-I both from healthy controls and ACS subjects.

Response: We would like to thank the Reviewer for the time spent for this evaluation and for the significant suggestions. The modifications made in the manuscript were highlighted in the corrected version of the manuscript. We agree with the Reviewer’s point of view, our results are not conclusive about the capacity of our test to predict the presence of ACS, mainly because the included patients were in the acute phase clinical manifestations. During this period, there are structural modifications of HDL that may affect the lipoproteins stability. Therefore, we were able to discriminate patients with coronary syndrome during the acute phase from control subjects. This is an important limitation of the study, and we have recognized it in the last paragraph of the Discussion section, page 9, as follows:

Moreover, patients were in the acute phase of the ACS; during this period, there are important alterations of HDL structure [3,4,19-21] that may have altered the stability of HDL.”

We also included the references 19 and 20 to emphasize this concept.In addition, we have also highlighted in the title of the corrected version of the manuscript the condition of acute phase in the patients.Concerning the HDL-cholesterol plasma concentration, we included patients within a wide range, from 17.9 up to 71.3 mg/dL. Of course, the real usefulness of this test to predict the incidence of coronary artery disease should be tested in future prospective studies. This expectative was stated at the end of the abstract and in the discussion section.We hope that the Reviewer finds that these changes respond satisfactory his/her concerns about our conclusions.

Specific comments:

Point 1. What is the effect of cholesterol (or phospholipids and other lipids) levels and apoA-I (or apoA-II and other protein) levels in HDL denaturation and stability?

Response: We thank the Reviewer for this question that help us to improve our results. To answer his/her question, we performed the additional determinations of apo A-I, HDL-triglycerides and HDL-phospholipidst hat were included in Table 1. As expected, apo A-I and HDL-phospholipids plasma concentrations were lower in ACS patients than in controls indicating a lower number of HDL particles. However, HDL-triglycerides were increased, suggesting that HDL particles were enriched with this lipid in patients. As an approach of the HDL lipid composition, we calculated the apo A-I-to-HDL-lipids ratio. These results indicated that the cholesterol and phospholipid content of HDL is comparable to HDL from control subjects. In contrast, the apo A-I-to-HDL-triglycerides ratio was about the half in patients with respect to HDL from controls, confirming an important enrichment of HDL particles with triglycerides.The Spearman’s correlation analysis(Table 2)showed important correlations of the percentage of HDL denaturation with the components of HDL, i.e. apo A-I, HDL-C and HDL-Tg. There was also an important correlation of HDL denaturation with the apo A-I/HDL-Tg ratio, indicating that triglycerides content is one of the most important determinants of the percentage of HDL denaturation. It is important to notice that the correlations were enhanced by the segregation of the groups in clusters; such segregation was the result of the huge difference of HDL denaturation between patients and controls, emphasizing again the importance of the new parameter that we describe in this study.Since patients, but not controls, were received statins,the correlations between the percentage of HDL denaturation and the components of the lipoprotein were lost when of statins intake was included as covariable in the analysis. However, the observed correlations should not be neglected because they apport important information about the possible structural features that contributes to HDL stability.

All these results have been included in Table 1, Table 2, and in page 7 of the corresponding section of the corrected version of the manuscript. We also mentioned in the Abstract some of these results and the importance of the HDL composition to the stability of the particle.

Point 2. Do HDL from ACS subjects with similar cholesterol levels as controls display difference in denaturation?

Response 2: In the first version of this manuscript, we postulated that HDL-cholesterol may be a surrogate of HDL stability. On this basis, at a same plasma HDL-cholesterol level, one could expect a similar percentage of HDL denaturation,as the Reviewer anticipated. To answer the query, we paired patients and controls by HDL-cholesterol at ±1mg/dL of difference. We were able to match9 patients with their corresponding controls.In fact, the difference of HDL denaturation between subgroups was as large as for the entire group. Therefore, at a same level of HDL-cholesterol, the percentage of HDL denaturation remain is still dependent of the ACS condition.

These results were included at the end of the corresponding section, page 8. We also included the Figure 5 to depict the paired controls and patients, and we briefly discussed these results in page 9.

Point 3. Was the plasma of ACS subjects collected during acute or stable phase?

Response: Yes, patients were in the hospital and samples were drawn within the 24h after the beginning of the symptoms, still during the acute phase of the coronary syndrome.This is a very important aspect that was not sufficiently emphasized in the previous version of the manuscript. In the new version of the manuscript, we modified the title to make clear the condition of acute phase in our patients. In section 4.1, Study population, we mentioned when the samples were drawn. The relevance of the acute phase on HDL structure was also remarked in the last paragraph of the discussion section, including additional references19 and 20, as follows:

Moreover, patients were in the acute phase of the coronary syndrome; during this period, there are important alterations of HDL structure [3,4,19-21] that may have altered the stability of HDL.

Finally, since acute phase could affect the stability of HDL, were cognized this condition as a weakness of our study as indicated in the answer to the General comments.

Round 2

Reviewer 2 Report

The authors have satisfactorily addressed my comments. The manuscript addresses the limitations of the study.